# The Early Global Vocation of Rome. Worship, Culture and Beyond

Anna Laura Palazzo 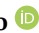

Department of Architecture, Roma Tre University of Rome, 00144 Rome, Italy; annalaura.palazzo@uniroma3.it

**Abstract:** In all likelihood, Rome was the first global city, holding such primacy for around two thousand years since the time when the Empire built strong integration and interdependence relationships with the whole *oecumene*. Against the backdrop of long-term beliefs powered by the Papacy, this paper highlights the main features of the global Rome as the very core of Christianity and, after several disruptive events from the Early Renaissance onwards, as a main destination of the *Grand Tour*. Making use of primary and secondary literature sources as well as of a substantial iconography, the paper investigates the interplay between power strategies and urban morphology—permanence/change—through two main lenses: (i) the 'inertia' over time of the radiocentric pattern of the *Forma Urbis* citywide, according to the old saying *all roads lead to Rome*; and, (ii) the relentless reuse processes over built-up areas and sense-making dynamics coupling tangible and intangible assets. Accordingly, the *Città Antica* and the *Città Moderna* would be intertwined in residents' and visitors' everyday experiences until the Age of Enlightenment, when a new sense of history was to require protection measures setting antiquities apart from city life. However, this is another story.

**Keywords:** Global Rome; *Forma Urbis*; *Città Antica*; *Città Moderna*; worship places; urban facilities; *Grand Tour*; urban metabolism





## 1. General Overview

### 1.1. Introduction

This contribution frames the lure of the Eternal City over nearly two thousand years, first as the capital of the Roman Empire and subsequently as the undisputed center of Christianity and primary destination of walking trails and pilgrimage routes from all over Europe. The well-known saying *all roads lead to Rome* was in fact first recorded in 1175 by Albanus ab Insulis, a French theologian and poet, whose *Liber Parabolarum* renders it as *mille viae ducunt homines per saecula Romam* (a thousand roads lead men forever to Rome).

According to methodologies and perspectives offered in the field of urban history, heritage studies and city planning, the paper interweaves two different readings of the interplay between permanence and change in the urban setting: (i) citywide, the material sedimentation over the *Forma Urbis Romae* dating back to antiquity (see Figure 1), that is, its radiocentric pattern—structure and layout—created by the consular roads (Castagnoli et al. 1958); and, (ii) the relentless reuse processes over built-up areas and sense-making dynamics coupling tangible and intangible assets (Boyer 1994; Roncayolo 2006; Germann and Schnell 2014; Battaglini 2019).

Over time, the ecclesiastical hierarchy prevailing over municipal institutions would carefully define and redefine the images and imagery of the Eternal City, drawing upon pre-existing traditions or shaping new ones in tune with current beliefs. Since written language was shared by a minority, oral traditions, typical of medieval and early modern societies, were prominent in providing guidance to the faithful. As a matter of fact, training for the clergy relied on repetition and memorization techniques of holy texts in Latin, generally neglecting the teaching of reading and ignoring the practice of writing. All over the Christian *oecumene* the establishment of decentralized territorial administrations—the

Dioceses under bishops' control—ensured widespread doctrine and liturgy circulation through 'parish schools'. Therefore, dominant orthodoxy would reach the remotest corners of the Christian universe, making it possible for anyone visiting Rome to directly appropriate an experience indirectly acquired (Le Goff 1957).

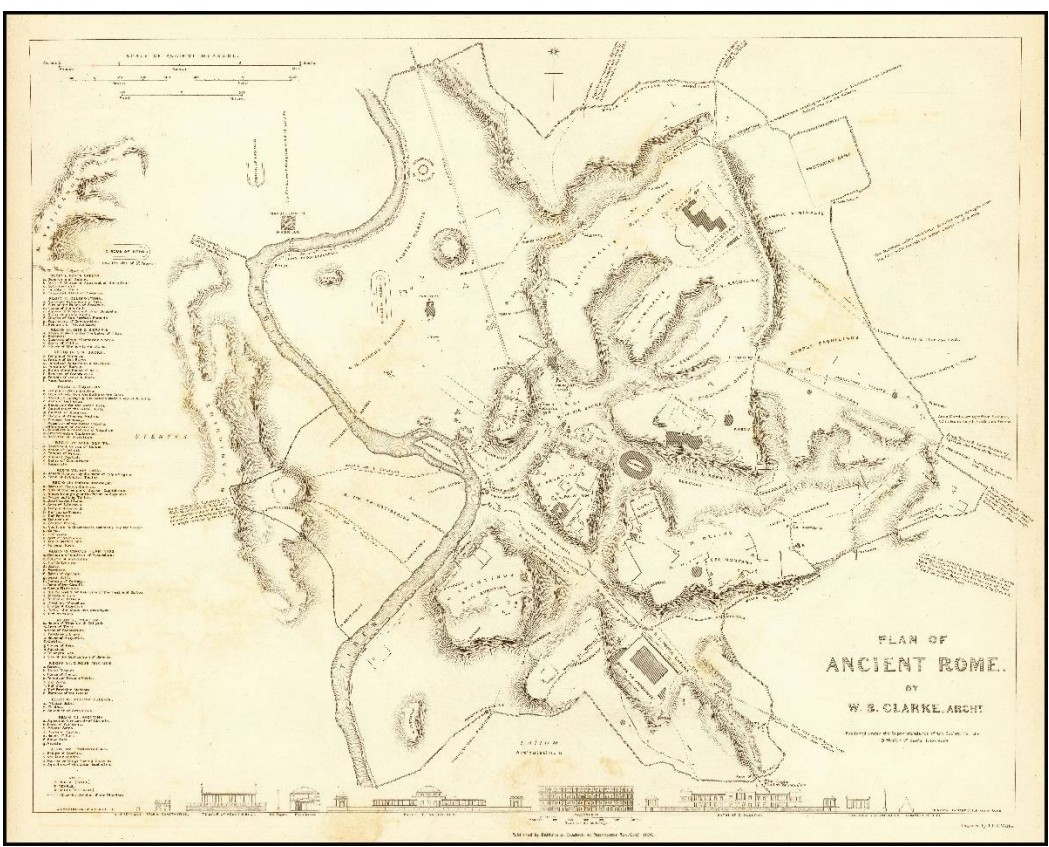

**Figure 1.** All roads lead to Rome. Plan of Ancient Rome, by W.B. Clarke, archt. Published under the superintendence of the Society for the Diffusion of Useful Knowledge. Engraved by J. & C. Walker. Published by Baldwin & Cradock, 47 Paternoster Row, 1 October 1830. (London: Chapman & Hall, 1844). Source: David Rumsey Collection. This work is licensed under a Creative Commons License. Digital images and descriptive data © 2021 by Cartography Associates.

The selection of pictures illustrating this contribution is intended to convey several recurring themes featuring and somehow bridging the physical and the symbolic cityscape of Rome through images which the audience was either familiar with or prepared to assimilate. As for the early phase of Christianity and the Middle Ages, seminal works have highlighted the heterogeneity of a culture within its paradoxical unity, allowing for 'a mosaic of opinions and sometimes contradictory behaviors yet flexible or vague enough not to hinder the freedom of speech or the effectiveness of practices' (Zumthor 1993). As for the Renaissance and Baroque period, power-led representations were increasingly challenged by social criticism and artistic research pointing out fragmentation in mindsets and beliefs foreshadowing modern anxiety (Starobinski 1964; Tafuri 1980).

In the beginning, Rome felt obliged to welcome pilgrims and visitors by accommodating worship places and basic facilities within the urban fabric. Accordingly, the Christian city would borrow from the Pagan one its references, memories, and symbols, with no need to make use of big gestures, except for the basilicas; placed next to the city walls and stretching up to the sky, they were erected to gather the Christian community for religious celebrations.

Over a thousand years later, with the *Renovatio Urbis Romae* under Sixtus IV (1471–1484), the Papacy would explicitly initiate imposing restoration and urban renewal activities, such as tracing new roads and entire neighborhoods for everlasting memory. In turn, urban

elites conformed to such customs, establishing or consolidating their social status and influence (Keyvanian 2019). From the 16th century onwards, due to compelling religious (Luther's Reform) as well as mundane stances, the primacy of spiritual and temporal power of the Papacy was severely challenged, and curiosity and inquiry would replace devotion in motivating travel. Rome was to become an unavoidable destination within the *Grand Tour* vogue that spread across Europe, a must for anyone who wished to be introduced to the culture and civilization radiating from the Eternal City. In response, Counter-Reformation hastened the pace of change citywide, reflecting its absolutist program in urban infrastructure and facilities that would awe residents and visitors alike (Giedion 1954).

The Age of Enlightenment has been established as the terminus *post quem* in this investigation; by that time, the Papal States were pressed internally by a huge secularization process. Concurrently, a new sense of history would require protection measures setting antiquities apart from everyday life.

### 1.2. Materials and Methods

This paper draws upon remarkable primary and secondary literature sources. Among the former, the main references have been the *Itineraria* and *Mirabilia Urbis*, first inventories of urban assets dating back to the custom of religious pilgrimage, and, when travel became a cultural matter, official city guidebooks in tune with the papal ideology, along with intimate tributes that visitors from Northern Europe used to pay to the Eternal City through their *Journal de voyage*. As for secondary sources, seminal works by historians, geographers and urban planners have been selected in order to parallel the experience of the real Rome and the image of the ideal one, both powered by the city rulers. Essays and articles have deeply investigated the evolution of urban form(s) at the city level and locally, focusing on building typologies accommodating everchanging worship and hospitality needs, and inquiring the overlapping of the *Città Antica*, *Città Sacra*, and *Città Moderna* (Lanciani [1901] 1988; Brumback 1957; Benevolo 1971; Frutaz 1962; Quaroni 1976; Insolera 1980; Krautheimer 1980; Lugli 1997; Manieri Elia 1998; Delbeke and Morel 2012). Some of the authors specifically addressing the binomial continuity/change in beliefs and behaviors contend that the feeling of 'eternity', both conveyed by the power structure and urban heritage, could be held responsible for the anthropological traits of its inhabitants, notably their lack of proactive citizenship. Due to such peculiar climate of the opinion, major changes in habits and city practice would only be triggered from outside (Quaroni 1959; Insolera 1980; Curcio and Manieri Elia 1982; Gross 1990).

## 2. Results

The results, discussed in the next section, highlight two different long-lasting approaches towards the use and reuse of the city: (i) from early Christianity to the late Middle Ages: *urban metabolism*; (ii) from the Renaissance to the Age of Enlightenment: *city of worship or city of culture?*

Unlike other European cities able to settle, at the turn of the first millennium, their cultural and economic life on new bases, establishing cathedrals of faith and municipality towers, Rome's population growth and development stagnated until the Modern Age. For a thousand years, the city enclosed by the Aurelian walls must have looked like a huge repository of building materials dotted by magnificent ruins among large stretches of wilderness (Duby 1966). In the early 15th century, some travelers' drawing pads captured such peculiar features of Rome so different from all other cities, in tune with the inherent sense of fate of its dwellers—indolence, indifference—featuring their enduring peculiar attitude that turned into art of survival (Frutaz 1962; Insolera 1980). There could be no greater contradiction between such drawings and sketches—hovels, nettles, ruins, and dusty poverty—and the everlasting outward iconography of the city (Huelsen [1907] 2016; Conti 2003). The disruption brought about by the Protestant Reformation to the core of Christianity would accelerate the pace of modernization started by Pope Sixtus IV. Notably, Pope Sixtus V (1585–1590) would display in a short timelapse an all-pervasive strategy

affecting the *Forma Urbis* as a whole, tracing and paving a new road system within the Aurelian walls while implementing water supply and water disposal networks; a modern city in its own right would be raised stating Catholicism's predominance and centrality.

## 3. Discussion

Section 3.1 investigates the ways the Papacy and its establishment have been addressing, for almost a millennium, the key issue of appropriating and taming tangible and intangible assets of the city's huge repository by subtle assimilation mechanisms of previous worship practices, simply shaping and molding existing building typologies in order to meet new needs and convey new values (Section 3.1.1). Such metamorphoses perfectly suited the rank of a city struggling with its declining power and oriented to revive as the world's primary spiritual center radiating its influence well beyond the nearest provinces. From this standpoint, in terms of global attraction at a distance, the *Mirabilia* and *Itineraria* provided visitors with an overwhelming array of narratives combining natural and super-natural aspects of the Christian (and Pagan) experience (Section 3.1.2), whereas hospitality structures and facilities embedded in the urban fabric brought relief to the sick, the poor, and the pilgrims as well (Section 3.1.3). Section 3.1.4 contends that, despite the shrinking of its urban population, the idea of perfection embodied by Rome was propagated in the dark ages and beyond by descriptions and images depicting its *Forma Urbis* citywide; a circular shape encompassed by the Aurelian walls centered on the *Miliarium Aureum* placed in the Forum by Emperor Augustus.

Section 3.2 moves into the Modern Age. In 1517, the Augustinian priest and university professor Martin Luther declared that the Papacy was no longer a source of truth. All over Europe intellectuals engaged in the journey to Rome in search of antiquities as a main source of inspiration, calling for safeguard measures (Section 3.2.1). Concurrently, two literary genres would see the light: (i) travel guides both for pleasure and religious tourism officially acknowledged by Counter-Reformation; (ii) the *Journal de voyage* recording visitors' personal impressions and reflections upon the transience of glory from a merely secular perspective. Meanwhile, the urban scene, heavily rearranged under Pope Sixtus V, would serve as a setting for new rituals and celebrations with great opulence (Section 3.2.2). The investigation ends at the turn of the 19th century, that brought about new reflections over the indivisible values of culture and civilization as the main components of *the real Museum of Rome* (Pinelli and Scolaro 1989). As a matter of fact, after the Restoration of the Papacy in 1814, a comprehensive set of measures addressing cultural heritage was promoted (Section 3.2.3). Quite explicitly, the Edict issued in 1820 by Cardinal Bartolomeo Pacca, alongside the usual concerns about the stability and decorum of Roman ruins, stated the importance of focusing on perceptive and functional relationships within their surroundings.

### 3.1. Urban Metabolism

3.1.1. Themes and Places

Soon after the Edict of Constantine granting freedom of worship for Christians (313 AD), the Empire proclaimed itself as the fulfillment of Augustus' *Pax Romana* as well as the Kingdom of God on Earth. According to Christian poet Prudentius: (4th century):

> all mankind came under the rule of the City of Rome to see the entire world linked by a common bond in the name of Christ. Grant then, Christ, to your Romans, a Christian City, a capital Christian like the rest of the world. Peter and Paul shall drive out Jupiter. (Krautheimer 1980, p. 42)

Since the early Middle Ages, at least once in a lifetime any good Christian was expected to journey to Rome. While pilgrimage practices reflected inner repentance paths, religious zeal and aesthetic admiration appeared to be one and the same, with no distinction between the sacred and the profane (Le Goff 1974; Le Goff 1987). Rome was soon confronted with the need to provide facilities both to local communities and pilgrims. However, the mission was far more subtle: the point was to commit pervasively and permanently the Pagan city, its shrines, and holy places, to Christian worship. Continuity in use would make it

easier to metabolize previous heritage. As regards intangible assets, the name of the saint celebrated in a church formerly serving as a Pagan temple had often a direct assonance with the god or hero celebrated in those same walls (Manieri Elia 1989). As for religious events, the Church Fathers tamed the main Pagan recurrences that drew legitimacy from centuries-old practices and marked the annual cycles that impacted on human activities. In fact, the dates of Christmas and Epiphany would replace previous important Pagan festivals. Replacing the feast of *Natalis Solis* which fell on December 25, Christmas Day was soon confirmed by new rituals. In turn, the festivity of the Epiphany, coinciding with solemn celebrations in honor of the sun and god Dionysus taking place in Syria and Egypt, was eventually consecrated as a main festivity by the Roman church. Urban space would be gradually appropriated by devotional practices; on Christmas Eve, a processional walkway wound from the Lateran, home to the pope, to Saint Peter's, crossing the city from east to west. The pope celebrated the first Christmas mass at St. Mary Major's, the second one at St. Anastasia's, as a tribute to the Byzantine court settled nearby, and the last one at St. Peter's.

Whereas 'for the pilgrims the psychological essence of the Christian journey resided in the reality hidden behind appearance' (Le Goff 1974, p. 193), the Papacy was interested in strengthening its global power and religious influence, even more so after the fall of Jerusalem in the hands of Saladin's Arab army (1187). Therefore, as Purgatory became more prominent in Christian thought, the idea took hold that during a person's lifetime it was possible to obtain a partial or total remission of sins by following the practice of fasting, praying, and almsgiving (indulgence). Lastly, the establishment of the Holy Years (with a periodicity of 100 and then 25 years) under Pope Boniface VIII (1294–1303) would grant the faithful visiting the Holy Seat a period of forgiveness.

Over time, the practice of selling indulgences spread as 'a way to reduce the amount of punishment one has to undergo for sins', providing a substantial source of wealth for the three industries still in place in Rome. For the first one, tourism, with taverns, overnight inns, and catering services, and the second one, construction, written sources provide evidence of an increase after the year 1000, while the third industry, represented by an elephantine bureaucracy, survived as a main legacy of the Roman Empire. Even though the city changed its appearance and meaning, it relied on centuries-old assets and infrastructure.

### 3.1.2. Itineraria and Mirabilia: The First Inventories of Urban Assets

Under Emperor Augustus, the Roman population had increased to around one million inhabitants, reaching its peak in the Antonine period (2nd century AD), with around 1,500,000 residents. Eventually, the city faced a long decline: due to recurrent turmoil at the boundaries of the Empire, a string of military crises required emperors to spend long periods far away, compelling Diocletian (emperor from 284 to 305) to establish the Tetrarchy, a system whereby four men ruled the Empire as a group. As a matter of fact, already from the late antiquity (3rd century), emperors were seldom natives of Rome, or even of Italy, and probably felt no particular affinity with the city. None of them based themselves in Rome, preferring cities closer to the imperial borders and potential trouble spots. Emperor Constantius II, son of Constantine and based in Constantinople, came to Rome on an official visit in 357:

> But though by Constantius II's day Rome had long since ceased to be the political, administrative, and strategic heart of the Roman Empire, it undoubtedly remained its symbolic center, the revered mother city from which the empire had grown. For this reason, a visit to the city by an emperor was a significant event. It was designed to convey to his subjects the emperor's power and lofty status. (Leveritt 2017)

The Emperor's tour included the Roman Forum, 'dazzling with its parade of wonders', the Colosseum and the Capitol with the temple of Jupiter, the Pantheon 'as big as an entire district of circular shape', the memorial columns of Trajan and Marcus Aurelius, the temple of Venus and Rome and the *Forum Pacis* by Vespasian, the theater of Pompey, the stadium

of Domitian and the Circus Maximus, where the Emperor attended the games and had an Egyptian obelisk raised. The visit ended in the forum of Trajan, where, in front of the equestrian statue of the Emperor, a Persian prince of the entourage told Constantius that 'if he too wanted a horse of that size, he would have to build an adequate stable'. (Krautheimer 1980, pp. 49–50)

This description would constitute an enduring model for two literary genres throughout the dark ages: the *Itineraria* and the *Mirabilia Urbis*.

The *Itineraria* were descriptions of the so-called *Viae Romeae*, formerly the consular roads, along with the main stages of pilgrimages from the northern regions, portrayed in full since the first half of the 8th century in the *Itinerarium Sancti Wiligelmi*. The *Itinerarium Einsidlense* was conceived to provide visitors with a rough description of the monuments along the roads 'entering the city from its twelve gates' and the transcription of several epigraphs. By the end of the 10th century, the Archbishop of Canterbury Sigeric the Serious journeyed to Rome along the *Francigena Way* to receive his pallium from the pope. He recorded his route and stops on the return journey consisting of 80 stages, averaging about 20 km a day.

In the 12th century, the *Mirabilia* established themselves as accounts of ancient monuments and ruins interwoven with anecdotal episodes. They enjoyed great popularity all over Europe and were translated into Romance languages. The *Liber Polypticus* (1142–1143) ordered by future Pope Celestine II to the Vatican Canon Benedict includes a list of

> city walls; gateways; the hills of Rome; bridges; buildings; triumphal arches; baths; theaters; cemeteries; memorial columns; the places of the passion of the Saints; the Vatican and the obelisk; the pine cone that was in Rome; temples; the temple of Mars; the Capitol; the Colosseum; the castle of Crescentius; the vision of the emperor Octavian and the response of the Sibyl; the marble castles in Rome; why the Pantheon was erected; why Octavian was called Augustus, which shrines are in Transtiberim. (Krautheimer 1980, p. 249)

### 3.1.3. Worship and Hospitality Assets

During the Christian persecution, liturgical celebrations ending with the blessing of the bread and the wine were secretly held in private homes or community centers (*tituli*), generally bearing the name of the owner of the *domus*. In some cases, when eventually the ancient *tituli* became churches, this transition occurred quite gently; the former name was retained, and the owner of the house was held as a saint.

By that time, the cult of martyrdom was widespread and led to the adaptation of previous *heroa*, round structures erected following the pagan tradition of the memory of legendary figures, or to the construction of analogous circular *martyria*. According to the pagan custom, the former Christian holy places were located outside the city, along the consular roads (see Figures 2 and 3). It is the very case of the cemetery of St. Lawrence on the Tiburtina Way, and of the one of St. Sebastian on the Appian Way, dedicated to the memory of the apostles Peter and Paul (known in earlier times as *Basilica Apostolorum*).

Refurbishment of the martyrs' burial places was often realized by merging elements of the *basilica* (a rectangular walled structure reserved for public purposes) and the *martyrion*—which would become an ambulatory around the tomb of the saint—obtaining the so-called *aula absidata*. By the 6th century, based on this layout, the main basilicas—notably the one devoted to the Virgin Mary and the shrines hosting the tombs of Saint John, Saint Peter, Saint Paul, and Saint Lawrence—had been erected to gather the multitude of faithful and let them move along the aisles and around the apses.

In the beginning, Christians did not care about the destiny of their mortal remains, nor did they seem to differentiate between a burial in a *colombarium* necropolis and a private tomb. Later, when beliefs about the resurrection became consistent, burial practice supplanted cremation, urging the construction of galleries and burial chambers—so-called catacombs—arranged on several levels to use all the available space. At the same time, devotional practices based on the corporeality would foster the cult of relics. New burial

grounds would be gradually installed in front of the churches, or in the *circuitus ecclesiae*, that is, in the space surrounding the walls and arches of the shrine. The spread of interment practices would also spur the creation of enclosures provided with a roof, dating back to the pagan tradition—the *coemeteria subteglata* (*coemeterium* means sleeping place in Greek)— next to the tomb of the saint ritually celebrated (*ad Sanctum*). Alternatively, graves were arranged around the apse of covered cemeteries, notably *sub stillicidio*, where percolating water from the sloping roofs would bless the faithful who had fallen asleep waiting to be called to heaven.

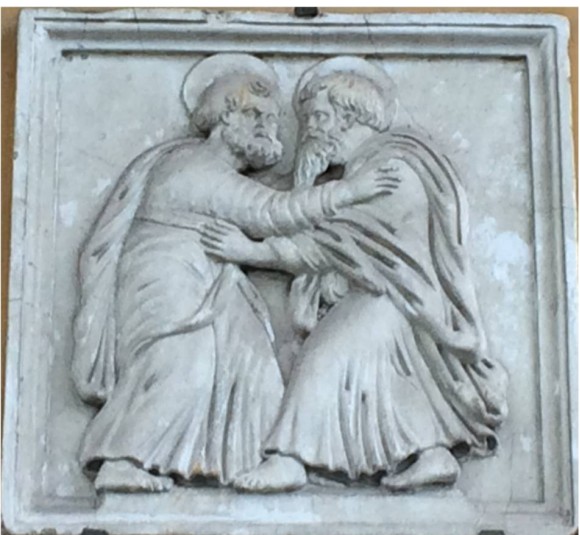

**Figure 2.** The apostles Peter and Paul greet each other before being martyred: bas-relief along the Ostiense Way (image by the Author).

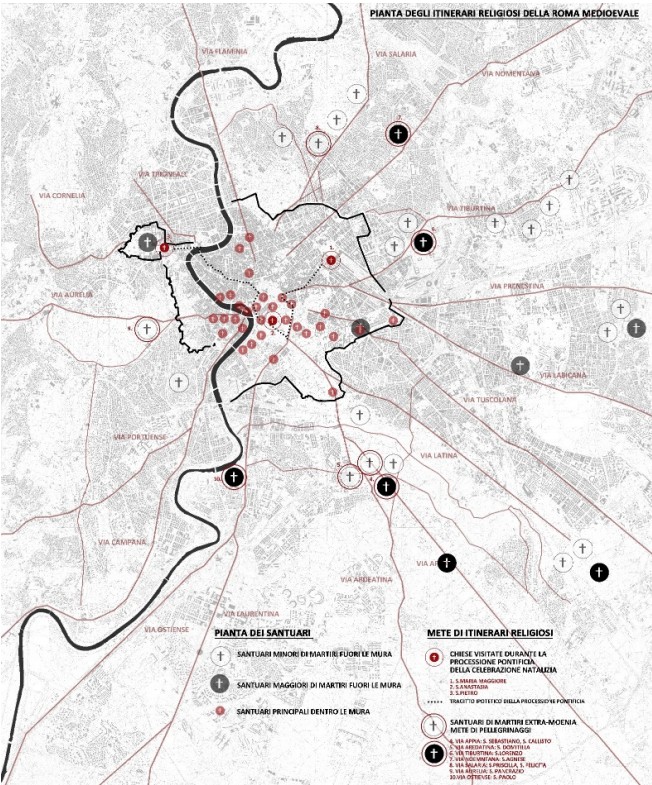

**Figure 3.** Religious destinations in the city and outside the walls along the consular roads (Author's work based on data from Krautheimer 1980).

The Council of Braga (563 AD) would ban burials inside churches, allowing graves to be placed along their perimeter walls. However, throughout the Middle Ages, cemeteries were crowded public places within the city and a main pilgrimage destination, often providing asylum for varied humanity seeking protection against secular justice; in their consecrated space, fugitives could enjoy impunity, while religious ceremonies mingled with fairs and markets, games, dances, and assemblies.

As for hospitality to the faithful, since the Edict of Constantine, facilities were provided to the poor, the sick, and the pilgrims. In the beginning, all kinds of shelters could provide various services to everyone in need, with the exception of the *Xenodochia*, that according to their Greek etymological root, were the first guesthouses. Subsequently, the term *Xenodochium* disappeared in favor of terms such as *Hospital* or *Hospitium* previously used to indicate the hospitality to the poor; these facilities seldom claimed remuneration. Other facilities, among which *Pandocheia*, *Tabernae*, and inns, supplied food and a shelter overnight for a fee (Krautheimer 1980).

In the early Middle Ages, the city lost substantial population and the few remaining inhabitants moved to the valleys of Forum and Campus Martius, where despite the Goths having cut off the water supply (6th century), they could access the Tiber's banks quite easily. Since in this area the original ground level was the same as the current one, the core of Ancient Rome, albeit seamlessly reshaped by everchanging needs, remained lively and busy for centuries.

Soon after the fall of the Roman Empire (476 AD), so-called *Diaconiae*, filiations of the papal administration, were established in order to provide assistance and relief to local population and refugees from the regions occupied by the Lombards. Run by monastic communities under the direction of a layman, the *Pater Diaconiae*, and supported by the rents of the *Patrimonium Petri,* such institutions dealt with food distribution to the local population and were strategically located next to the markets of the Pagan city, such as the Port of Ripa Grande, the Foro Olitorio, and Foro Boario, the most crowded places in the city. Often housed in existing buildings, the *Diaconiae* were barely noticed in the urban fabric, fully participating in the assimilation and domestication strategy of the Pagan remains.

Despite its decline, Rome kept welcoming small communities of foreigners—notably Greeks, Saxons, and Slavs—who settled along the Tiber performing trade, handcrafts, and river-based activities. Their neighborhoods were provided with own facilities, called *Scholae*, accommodating religious and care centers (Calabi and Lanaro 1998).

Although historical sources are silent about the real size of the city in the dark ages, it appears all too clear that barbarian invasions weakened urban defenses and assistance, forcing the population to flee by organizing themselves in scattered settlements where it would be easier survive and obtain food (Vauchez 2001). The *Diaconiae* themselves soon fell into neglect.

It was only after the first millennium that all over Europe urban settlements would be triggered again by the renewed mobility of people and goods and recovery of monetary systems. Care facilities spread along the *Viae Romeae* thanks to Canons living in communities under a rule, and subsequently to knights hospitaller orders such as the Templars and the Jerusalemites providing protection to pilgrims in peacetime.

### 3.1.4. The Persistent Legacy of Rome Iconography

Following the description provided by the *Itinerarium Einsidlense*, German scholar Christian Huelsen drafted a hypothetical map of Rome in the 8th century, in a circular shape, whose center in the *Forum* was named *Umbilicus Urbis* (Huelsen [1907] 2016). As a matter of fact, the same monuments, churches, and other city landmarks listed in the *Itinerarium* and, quite strikingly, in centuries-old *Res gestae* written by historian Ammiano Marcellino who reported the visit of Emperor Constantius II, are sketched in a precious miniature adorning the prayer book of the Duke of Berry, depicted as a fairy tale illustration celebrating the wonders of Rome. In line with a flourishing imagery encompassing enduring values of sacred and profane antiquity strongly imbued with symbolic meanings, it portrays the city walls and gates, the aqueduct of Nero, the Colosseum and Porta Maggiore, the memorial

columns of Trajan and Marcus Aurelius and the statue of the latter, and the twin statues of
the Dioscuri. In addition, other monuments are depicted, such as the Castle of the Holy
Angel, St. Peter's, St. Paul's outside the walls, the basilica of St. John in Lateran, the Holy
Cross, the Colosseum, the Capitol, the monument of Augustus, St. Mary's in Cosmedin, St.
Cecily's, the Anguillara tower, the Pyramid, St. John's and Paul's, the Claudius Aqueduct,
St. Bibiana's, and Castro Pretorio barracks. Such iconography, recording the twelve gates
in the Aurelian walls while omitting the road system, proves faithful in unfolding the
overriding rule of the *Forma Urbis* until our day. The circular shape also lends itself to the
hypothetical reconstruction of the 'Ancient Capitolium' by cartographer Sebastian Munster
one century later (see Figures 4 and 5).

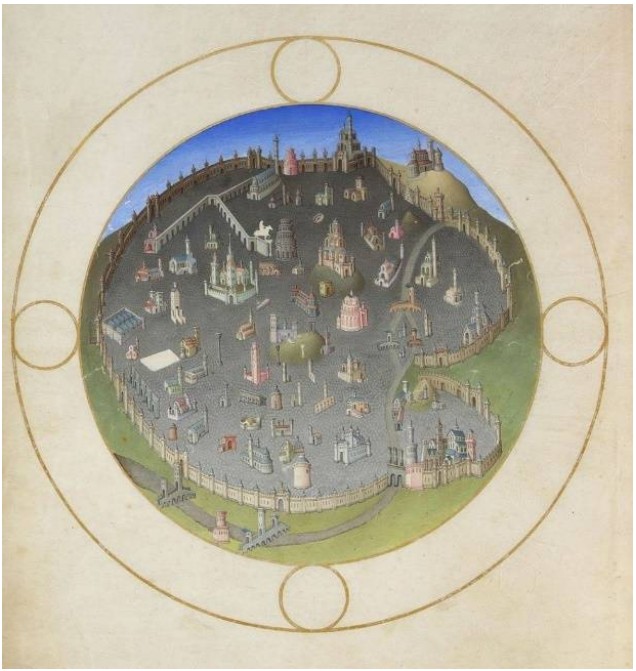

**Figure 4.** Limbourg Brothers, *Très Riches Heures du Duc de Berry*, early 15th century (image by the
Author). The coeval Map of Rome by Taddeo di Bartolo, a fresco in Palazzo Pubblico in Siena, shows
the same perspective and monuments.

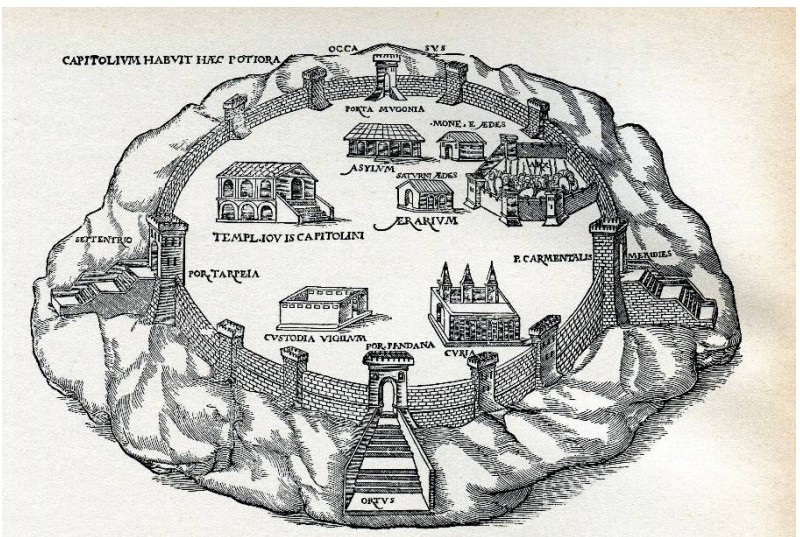

**Figure 5.** Sebastian Munster, *Cosmographia universalis*, 1544. The *topos* of the closed circular shape
also applies to the *Capitolium*, the core of ancient Rome. Source: *Urbanistica* 1959, 27, p. 6.

Meanwhile, the real city was far from an accomplished masterpiece. After the displacement of the pope and curial officials to Avignon and a great plague (1348), the population fell to about 17,000 inhabitants, gradually increasing to some 60,000 at the beginning of the 16th century (Insolera 1980; Smith and Gadeyne 2013).

*3.2. City of Worship or City of Culture?*

3.2.1. The Cultural Shift

From the Renaissance onwards (1500–1600), a shift in mindsets would prompt new habits and lifestyles unconstrained by both the overriding principle of religious authority and the casual reuse of urban fabric. This transition can be understood by three leading thinkers' inspiring words: Martin Luther, calling back to the sources of Christianity, Raphael Sanzio, supporting the revival of the Roman civilization, and Michel de Montaigne, connecting the lure of the Eternal City and its actual role in the knowledge of antiquity and bringing pleasure and curiosity to the journey experience.

The scandal of the trading of indulgences burst abruptly with the accusation of Martin Luther, German professor of theology, deeply struck by corruption, cynicism, and immorality caught among the clergy during his stay in Rome (1510):

> I would not take 100,000 florins not to have seen Rome, although I do not yet thoroughly know its great and scandalous abominations. When I first saw it, I fell to the ground, lifted up my hands and said–Hail, thou holy Rome, yes, truly holy, through the holy martyrs, and their blood that has been shed there. [ . . . ] Nobody would believe, unless he saw with his own eyes the licentiousness, the vice and the shame that is in vogue in Rome. [ . . . ] As was my case in Rome, where I too, was a mad saint, ran the round of all the churches and vaults, and believed every lie that was invented there. (Brumback 1957)

Luther argued that salvation is not earned by good deeds but only received as a gift from God. His theology challenged the authority of the Pope, claiming that the Bible is the only source of divinely revealed knowledge and rejecting the mediating role of the Church and its priestly order vis-à-vis the faithful.

The Letter to Pope Leo X written in 1514 by Raphael in his capacity as Superintendent of Roman Antiquities sounds as a proud claim of the individual skills and artistic qualities of an intellectual elite that was formed in connection with a new approach to history:

> Therefore, Holy Father, let it not be the lowest of Your Holiness's priorities to ensure that–out of respect to those divine spirits, the remembrance of whom encourages and incites to virtue the intellects among us today–what little remains of this ancient mother of the glory and renown of Italy is not to be completely destroyed and ruined by the wicked and the ignorant. Unfortunately, even here these people have perpetrated evil deeds against those souls who, with their blood, brought so much glory to the word, to this state and to us. Rather, by preserving the example of the ancients, may Your Holiness seek to equal and better them, as indeed you have done through your magnificent buildings, by supporting and favoring the virtues, reawakening genius, rewarding virtuous endeavors, and by sowing that most holy seed of peace among Christian princes. (Hart and Hicks 2009)

As the Church saw its spiritual power reduced, Rome entered a new phase, tied to its appeal as a city of art and history besides and beyond religious motivations. The journey to Rome, inaugurating the cult phenomenon of the *Grand Tour* and a literary genre, would be increasingly devoted to the search for humanistic sources. Since the golden age of Elizabethan England, the *Grand Tour* was deemed a must for educating and enabling young gentlemen to occupy their place in the world. Contemplation of memories would be one with the desire for collecting ancient artefacts and antiquities and to be portrayed on a backdrop of monumental ruins (Brizzi 1976; De Seta 1992; Brilli 1995; Brilli 2008; Leibetseder 2017). French and German aristocracies were also captured by this attitude.

Michel de Montaigne, who was used to traveling for pleasure and instruction, praised the journey as a fruitful exercise (de Montaigne [1592] 1965):

> The soul gets continuous excitement to notice the Unknown and the New. I do not know better training to life than proposing relentlessly the diversity of so many other lives, imaginations and uses and offering a perpetual variety of forms of our nature. My body is neither idle nor worn, and such agitation rushes it. I ride a horse without being bored for eight and even ten hours a day.

The Papacy had reacted to the Protestant Reformation by endowing the city with new devotional routes and a spectacular array of rituals. The *Itinerari Filippini*, somehow related to the primal tradition of the pontifical liturgies kept alive until the Avignon Papacy (14th century), owe their revival to St. Filippo Neri who managed to allow annual plenary indulgence through a pilgrimage along the so-called 'Seven Churches Walk'. These *stationes*, corresponding to the main holy places at the time, were touched in a repentance path heading to the catacombs along the Appian and Ardeatina Ways (1552). Filippo's very innovation lies in making the visit a collective practice with rest and recreational moments in the urban spaces and churchyards. The devotional path wound from St. Peter's, through St. Paul's, St. Sebastian's, St John's, the Holy Cross, St. Lawrence's, and St. Mary Major's. Given the length of the itinerary (20 km), it was traveled in two days. The first one was devoted to St. Peter's and the second one to the other six basilicas.

As for the *Itinerari Giubilari*, during his short pontificate, Sixtus V gave life to a citywide polycentric layout to be experienced sequentially (1585–1590). The Sixtine Urban Plan mirrors the power of the church and is the theater of its rituals and protocols, celebrated by the Pope in person; the new road system consists of straight axes visually and symbolically connecting holy places dotted by four great obelisks (at St. Peter and St. John's Squares, Piazza del Popolo, and Piazza Esquilino). This all-pervasive strategy is in tune with the renowned motto by St. Ignatius of Loyola, founder of the Society of Jesus order—*deformata reformare, reformata confirmare, confirmata renovare*—that applies to a four-week spiritual exercise training, where *forma*, in its different meanings denoted by different prefixes, refers to letting oneself be molded by the mystical experience (Manieri Elia 1989) (see Figures 6 and 7).

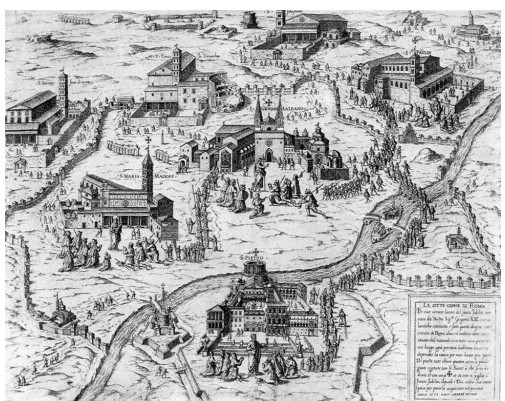

**Figure 6.** Antoine Lafréry. Pilgrims visiting the Seven Churches of Rome during the Holy Year of 1575. Source: Metropolitan Museum of Art. Available under the Creative Commons CC0 License.

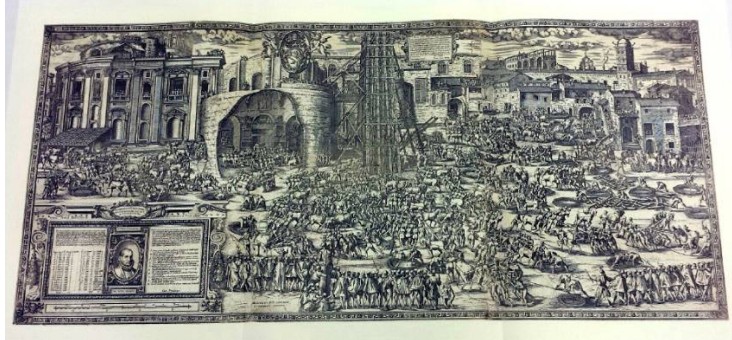

**Figure 7.** Domenico Fontana. Engraving adorning the book Della trasportatione dell'obelisco Vaticano et delle fabriche di nostro Signore Papa Sisto V, 1590 (private collection).

### 3.2.2. Rome Narratives: 'Official' Accounts and Personal Insights

The image of the city subtly disclosed by Counter-Reformation and spread around the world would intertwine the profane and the sacred—namely the ancient and the Christian city—with a narrative interweaving the visible and invisible. The new genre of the guidebook, acknowledged by the Papacy and richly illustrated with engravings, did not differ much from the hagiographic intentions of the *Mirabilia*. They usually contained a section on antiquities; a section on churches; a guided tour of the city; lists of popes, emperors, and indulgences; and a description of the marvels of the world. Iconography was provided by sophisticated printed images of antiquities, and the expanding book market was intended to meet the interest of ever larger categories of visitors and city users seeking the roots of the ancient city in the present one. Mediating between different needs, in the 17th century *Roma Moderna* is singled out as a new category encompassing contemporary developments, while *Roma Sacra* is the expression of the Christian city.

> The emergence of the modern city would destabilize the quite neat separation between *Roma Antica* and what was initially thought of as *Roma moderna* (or Christian Rome) in the *Mirabilia Urbis*. [ . . . ] The progressive emergence of contemporary Rome between the Christian and ancient city appeals to the interests of this category of visitors. Still, the permanence of the religious part of the guide until circa 1700, together with the fact that the contemporary city becomes present in the guidebook at the expense of ancient–not Christian–Rome, points to Rome's uniqueness. (Van Acker and Uyttenhove 2012, p. 399)

To the visitor, juxtapositions and overlaps among the Three Romes, displaying past glory and current misery, were a source of curiosity and dismay at once and a reason for reflections upon current decline and the neglect of things and people.

Gaetano Visconti Volonteri, traveling to Rome in the early 19th century, recorded his impressions in his diary:

> All in all, a lot of admiration combined with a lot of revulsion for the many beautiful things mingled with the sad ones arising everywhere. Such are the impressions that Rome inspires at first glance, combining with a feeling of indignation at the indolence, neglect, absolute inability of the inhabitants of Rome. Romans (are) brought up in idleness and leisure. Professions, trades, fine arts are left to foreign entrepreneurship. Every talent, every industry endeavor to deceive foreigners led by their curiosity to admire ancient and modern masterpieces within the City [ . . . ]. Under other standpoints, Rome opens to an immense field of beauty and curiosity.

In fact, for the 'pilgrims' of the Grand Tour, as Goethe used to refer to himself, Italy and Rome were the warm passionate south as opposed to the cautious north; the place where the classical past was still alive, although in ruins.

### 3.2.3. The Age of Enlightenment and Its Influence

> Every city has its history, but in Rome, the past is felt to exist in the present to a higher extent than in other, less culturally encoded cities, both as a practice established by cultural tradition, and as visually manifested in the urban space. (Blennow and Rota 2019, p. 8)

Over the 18th century, along with the mainstream approach to antiquity increasingly nourished by philological methods free from the Renaissance principle of emulation, new imaginary paths cast a light on the emerging spirit of the age. Giovanni Battista Piranesi (1720–1778) cultivated both approaches, as a prominent scholar of the classics, anticipating the scientific methods of modern archeology, as a well as a renowned engraver fascinated by the view of the *talking ruins* of the Eternal City. Piranesi acknowledged that the Papal States did not have the power and resources to preserve such a huge heritage, whose owners were unable to take forward-looking initiatives. The *Pianta di Roma disegnata colla situazione di tutti i Monumenti antichi* was inspired by the *Forma Urbis Severiana* engraved on marble slabs between 203 and 211 AD, originally located in the Temple of Peace. The fragments are out of context and out of scale, conveying a sense of anxiety and uncertainty (see Figure 8).

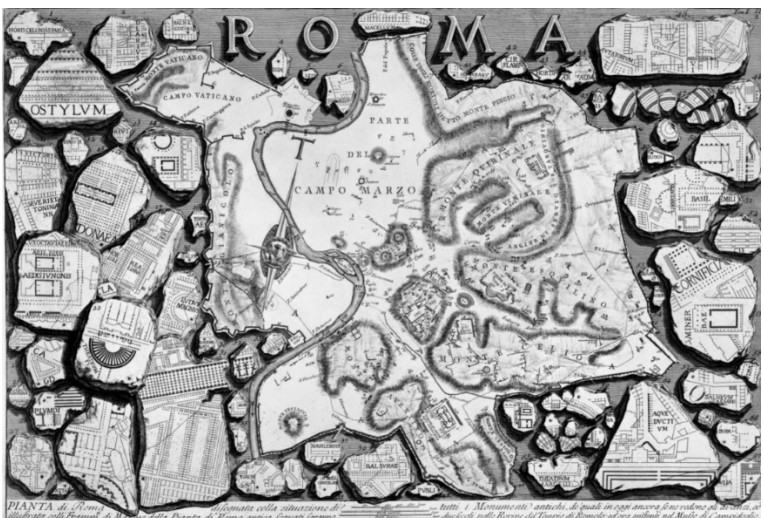

**Figure 8.** Giovanni Battista Piranesi, Pianta di Roma disegnata colla situazione di tutti i Monumenti antichi, de' quali in oggi ancora se ne vedono gli avanzi, ed/illustrata colli Frammenti di Marmo della Pianta di Roma antica, scavati, saranno due secoli, nelle Rovine del Tempio di Romolo; ed ora esistenti nel Museo di Campidoglio. This work is licensed under a Creative Commons License. Digital images and descriptive data © 2021 by Cartography Associates.

In the Age of Enlightenment, the centuries-old reflection over the destiny of the Ancient Rome, still the core of a busy city grown upon or alongside monuments and ruins, was confronted with new achievements addressing both the universal values of culture and their place-specific features (Fehér 1992). At the time, temples and porches, arches, and vaults were still an all-pervasive backdrop to everyday life and current activities in the city. The ancient Forum was used for grazing, as well as a meeting place for the people living nearby; the toponym *Campo Vaccino* derives from the cattle market that used to be held there, next to the Customs of the *Grascia* (animal fats for food consumption). Such picturesque beauty was about to be replaced by the largest and most important urban archaeological complex in the world, encompassing the Roman Forum and Palatine Hill (see Figures 9 and 10). Thanks to the ideas of A.C. Quatrémère de Quincy, who called for preservation measures in his passionate *Lettres sur le préjudice qu'occasionneroient aux Arts et à la Science le déplacement des Monuments de l'Art en Italie, le démembrement de ses Ecoles, et la*

*spoliation de ses Collections, Galéries, Musées, etc.*, the legacy of Ancient Rome was to deserve ever more attention among archeologists and scholars from other fields:

> The real Museum of Rome, the one I am speaking of, is made of statues, colossi, temples, obelisks, triumph columns, baths, circuses, amphitheaters, arches, tombs, stuccos, frescoes, bas-reliefs, inscriptions, fragments of ornaments, building materials, furniture, tools, etc. However, it is equally composed by places, sites, mountains, ancient roads, and their mutual relations within the ruined city, widespread memories, local traditions, still existing habits, connections and comparisons that can only be done on-site. (Pinelli and Scolaro 1989)

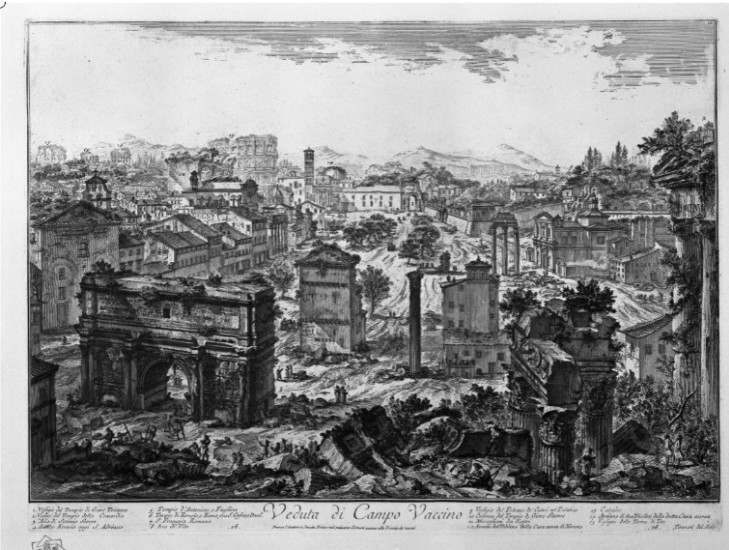

**Figure 9.** Giovanni Battista Piranesi, View of Campo Vaccino, 1748. Source: https://commons. wikimedia.org/w/index.php?title=File:Vedute_di_Roma_ (accessed on 20 April 2021).

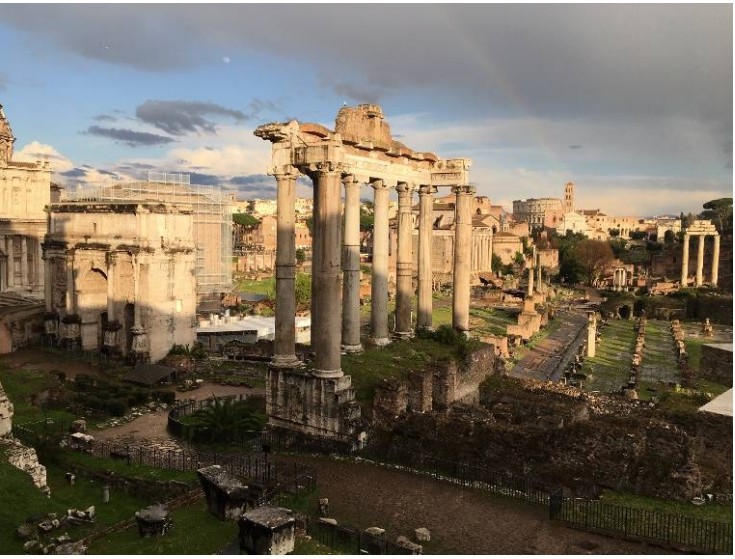

**Figure 10.** The Fori at present (image by the Author).

Despite social unrest, the short interlude of the Kingdom of Italy, and the Napoleonic occupation of Rome with the exile of Pius VII (1809–1813), the restoration of the Papacy would initiate a new phase in public affairs; notably, the Edict of Cardinal Pacca issued in 1820 would enforce proper documentation of monuments and above ground remains and make compulsory the declaration of accidental discoveries during excavation works (Emiliani 1978; Benevolo and Scoppola 1988).

After the unification of Italy in 1870, the whole Fori district would be placed under protection, becoming a large open-air museum of the Roman antiquities. The point was then to demolish urban fabric, displace inhabitants elsewhere, and disclose previous structures (Benevolo 1971; Insolera and Perego 1983; Tintori 1985; Palazzo 1993; Casini 2017).

## 4. Conclusions

Global Rome, as the capital of the Empire, the epicenter of Christianity, and a major destination of the Grand Tour, shaped both insiders' and outsiders' experience over time while fostering a huge literary production and iconography. The persistent lure of *Rome beyond Rome* draws upon different interpretation levels: ancient vs. modern; real vs. ideal; visible vs. invisible city. More specifically, two different although intertwined cities—the ancient and the modern one (the latter labeled as the Christian one)—would come to the fore. At the end of the 17th century, the *Città moderna* would obtain a space of its own thanks to the spectacular Baroque arrangements carried out by Pope Sixtus V, while the previous *strata* of the Christian city would become the *Città Sacra*.

This paper has discussed two different long-lived power-oriented strategies towards the use and reuse of the city.

In the first period, from early Christianity to the late Middle Ages, Rome kept pre-existing infrastructure and facilities, and the built environment underwent never-ending erasures and rewritings. In fact, the Christian city could graft onto the Pagan one, reappro-priating pre-existing structures and adapting urban fabric to new needs.

In the second period, the Counter-Reformation established a new pace of change in urban morphology, arranging large-scale transformations. Well-renowned architects and urban planners, such as Domenico Fontana, were committed to a strategy of displaying the magnificence of a brand-new urban setting rivaling other destinations Europe-wide (Fontana [1590] 2000).

Modernization, however, was initially spurred by outside instances rather than by inner conviction. Among the main causes for the epistemological turn occurring in the Renaissance, this essay has singled out three different attitudes relating to: (i) the claim for authenticity of faith without any mediation; (ii) the reference to the classical legacy reviving the Latin saying *Historia est magistra vitae*; (iii) curiosity and a genuine spirit of inquiry as a main travel motivation. Such disruptions have triggered or accelerated a 'culture of modernity', from which the physical Rome drew its impulse to change. Further investigation might clarify whether and to what extent such pioneering reflections by Roman and foreign intellectuals have shaped common knowledge and perceptions.

As for expert knowledge, over the last two hundred years, the Central Archaeological Area of Rome, along with a huge heritage spread citywide, has been set apart from everyday life. The Enlightenment intuition of a new sense of history based on discontinuity in city making goes far beyond the ideal identification with classical antiquity permeating Humanism and Renaissance. The codes of classicism are no longer considered as a living language to be revived, and the huge legacy from antiquity becomes a subject of inquiry rather than a main source of imitation. Trust in the traditional rules no longer being an obligation, history would become a 'guide for life' less literally but more ideally. Critically oriented awareness in dealing with heritage conservation issues would replace an overall feeling of a perfect consonance with the past (Palazzo and Pugliano 2015).

The future of the most important archaeological park in the world is currently a major theme of debate: for a century now, antiquity is being challenged by itself and by the projections of modernity through protection and management measures as well as planning approaches, with often disappointing results. After so many centuries of history, this legacy should enhance the *aura* of the place, giving residents and visitors alike a keen awareness of the present and welcoming the sense of the past in an allusive way.

**Funding:** This research received no external funding. It has been performed over several years, lastly in the frame of the HERILAND Program, Cultural Heritage and the Planning of European Landscapes, funded by the European Union's Horizon 2020 research and innovation program under

the Marie Sklodowska-Curie grant agreement No 813883. The fees for open access publishing have partly been covered by Heriland.

**Institutional Review Board Statement:** Not applicable.

**Informed Consent Statement:** Not applicable.

**Conflicts of Interest:** The author declares no conflict of interest.

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
