# Peer review of "The Early Global Vocation of Rome. Worship, Culture and Beyond"

_humanities, doi:10.3390/h10030103_

Round 1
Reviewer 1 Report
It is a very interesting question of study, how Rome could have preserved a role of a centre since Antiquity. The article tries to show some factors leading to Rome's central position first in Middle Ages, second in Early Modern History (from Renaissance to the Age of Enlightment). Although the author reveals some interesting facts, a coherent and lucid argumentation would help the reader to have a clear picture on Rome's importance during centuries.
The arguments or illustrations seem to be a bit hectic. I have some doubts that historical processes or changes should be attributed to strategies (lines 96, 106, 539 et passim).
Some minor remarks:
The statement, that "Almost none of them was native of Rome, or even of Italy, and probably felt no 184 peculiar affinity with the city." (lines 184-5), must be specified. Many emperors since Trajan were born in provinces.
AD is unnecessary in line 163.
To paragraph beginning with line 388 can be suggested: https://brewminate.com/the-grand-tour-educational-journeys-from-the-16th-to-18th-centuries/
It would be interesting to mention that the sentence (all roads lead to Rome, line 13) is from Albanus ab Insulis (12th century), who meant spiritual way. T
Author Response
I am extremely grateful to the Reviewer for his keen insight, remarks and suggestions.
All corrections and clarifications are in red colour in the attached pdf file.
The literature review and references have been extended also to theoretical contributions.
As for the main remark:
The arguments or illustrations seem to be a bit hectic. I have some doubts that historical processes or changes should be attributed to strategies (lines 96, 106, 539 et passim).
I hope I have better framed and detailed the main focus of my investigation in Paragraph 1.1 that has been completely re-written.
The selection of pictures illustrating the work displays some recurring themes featuring and somehow bridging the physical and the symbolic cityscape of Rome through images which coeval audience was acquainted to or prepared to assimilate, the more so in the Middle Ages, when the duty to convey Christian doctrine was entrusted to oral tradition.
Moreover, I revised the passages somehow mitigating the emphasis on the strategic dimension of urban interventions, however determined by the will to communicate values and influence insiders’ and ousiders’ perceptions of the city. In this respect, and according to the consulted literature, I clarified the hiatus that was created with the Renaissance between the early approach hinging on urban metabolism (urban morphologies are reinterpreted and tamed under a Christian perspective) and subsequent intentional logic to leave indelible traces in the urban fabric in tune with Papal absolutism.
As for minor remarks:
To paragraph beginning with line 388 can be suggested: https://brewminate.com/the-grand-tour-educational-journeys-from-the-16th-to-18th-centuries/
Thank you for pointing this out to me. This insight proved very useful.
It would be interesting to mention that the sentence (all roads lead to Rome, line 13) is from Albanus ab Insulis (12th century), who meant spiritual way.
Thank you for specifying the context of the saying. I devoted few words (in red) at the beginning of the contribution.
The statement, that "Almost none of them was native of Rome, or even of Italy, and probably felt no 184 peculiar affinity with the city." (lines 184-5), must be specified. Many emperors since Trajan were born in provinces.
Thank you, a specification was made (in red).
I am extremely grateful to the Reviewer for his keen insight, remarks and suggestions.
All corrections and clarifications are in red colour in the attached pdf file.
The literature review and references have been extended also to theoretical contributions.
As for the main remark:
The arguments or illustrations seem to be a bit hectic. I have some doubts that historical processes or changes should be attributed to strategies (lines 96, 106, 539 et passim).
I hope I have better framed and detailed the main focus of my investigation in Paragraph 1.1 that has been completely re-written.
The selection of pictures illustrating the work displays some recurring themes featuring and somehow bridging the physical and the symbolic cityscape of Rome through images which coeval audience was acquainted to or prepared to assimilate, the more so in the Middle Ages, when the duty to convey Christian doctrine was entrusted to oral tradition.
Moreover, I revised the passages somehow mitigating the emphasis on the strategic dimension of urban interventions, however determined by the will to communicate values and influence insiders’ and ousiders’ perceptions of the city. In this respect, and according to the consulted literature, I clarified the hiatus that was created with the Renaissance between the early approach hinging on urban metabolism (urban morphologies are reinterpreted and tamed under a Christian perspective) and subsequent intentional logic to leave indelible traces in the urban fabric in tune with Papal absolutism.
As for minor remarks:
To paragraph beginning with line 388 can be suggested: https://brewminate.com/the-grand-tour-educational-journeys-from-the-16th-to-18th-centuries/
Thank you for pointing this out to me. This insight proved very useful.
It would be interesting to mention that the sentence (all roads lead to Rome, line 13) is from Albanus ab Insulis (12th century), who meant spiritual way.
Thank you for specifying the context of the saying. I devoted few words (in red) at the beginning of the contribution.
The statement, that "Almost none of them was native of Rome, or even of Italy, and probably felt no 184 peculiar affinity with the city." (lines 184-5), must be specified. Many emperors since Trajan were born in provinces.
Thank you, a specification was made.
Reviewer 2 Report
The paper on the History of ancient Rome is an original one.
The quality and scientific soundness of the paper is very good.
Author Response
I am extremely grateful to the Reviewer for his keen insight and suggestions.
I hope I have better framed and detailed the main focus of my investigation in Paragraph 1.1 that has been completely re-written.
All corrections and clarifications are in red colour in the attached pdf file.
The literature review and references have been extended also to theoretical contributions.
Reviewer 3 Report
The topic of the post is very unique. It is a multidisciplinary connection of architecture, urban planning, history, theology, etc. An extensive historical excursion perfectly illustrates the development of the city's possibilities to provide the pilgrim with a religious character with what he comes for. The author describes the various stages of prosperity and decline of the city. He also deals with the technical side of burial in early Christians in the city of Rome. The author tries to describe the city in the sense of the main center of European Christianity. It also comments on the hypothetical appearance and character of the city during the Middle Ages. I appreciate the effort to capture the influence of intellectual currents since the Renaissance. St. Philip Neri and St. Ignatius of Loyola are highlighted, which are known facts. The study seeks to plastically capture the impact of the city of Rome on a visitor whose primary goal is a religious pilgrimage. It is a reflection on the historical use of the city. The study is well and richly referenced. I see no reason why, in the state in which it finds itself, it could not be published.
Author Response
I am extremely grateful to the Reviewer for his keen insight and suggestions.
I hope I have better framed and detailed the main focus of my investigation in Paragraph 1.1 that has been completely re-written.
All corrections and clarifications are in red colour in the attached pdf file.
The literature review and references have been extended also to theoretical contributions.
I have refined the spell check as requested.